# Reactive Species in Huntington Disease: Are They Really the Radicals You Want to Catch?

**DOI:** 10.3390/antiox9070577

**Published:** 2020-07-02

**Authors:** José Bono-Yagüe, Ana Pilar Gómez-Escribano, José María Millán, Rafael Pascual Vázquez-Manrique

**Affiliations:** 1Laboratory of Molecular, Cellular and Genomic Biomedicine, Instituto de Investigación Sanitaria La Fe, 46026 Valencia, Spain; jose_bono@iislafe.es (J.B.-Y.); ana_pilar_gomez@iislafe.es (A.P.G.-E.); millan_jos@gva.es (J.M.M.); 2Joint Unit for Rare Diseases IIS La Fe-CIPF, 46026 Valencia, Spain; 3Centro de Investigación Biomédica en Red de Enfermedades Raras (CIBERER), 28029 Madrid, Spain

**Keywords:** Huntington disease, huntingtin, neurodegeneration, oxidative stress, inflammation, microglia, astrocytes, free radicals, antioxidants, *C. elegans*, *Drosophila*, mouse models, clinical trials

## Abstract

Huntington disease (HD) is a neurodegenerative condition and one of the so-called rare or minority diseases, due to its low prevalence (affecting 1–10 of every 100,000 people in western countries). The causative gene, *HTT*, encodes huntingtin, a protein with a yet unknown function. Mutant huntingtin causes a range of phenotypes, including oxidative stress and the activation of microglia and astrocytes, which leads to chronic inflammation of the brain. Although substantial efforts have been made to find a cure for HD, there is currently no medical intervention able to stop or even delay progression of the disease. Among the many targets of therapeutic intervention, oxidative stress and inflammation have been extensively studied and some clinical trials have been promoted to target them. In the present work, we review the basic research on oxidative stress in HD and the strategies used to fight it. Many of the strategies to reduce the phenotypes associated with oxidative stress have produced positive results, yet no substantial functional recovery has been observed in animal models or patients with the disease. We discuss possible explanations for this and suggest potential ways to overcome it.

## 1. Introduction

Some neurodegenerative diseases, such as Parkinson’s disease (PD), amyotrophic lateral sclerosis (ALS), Alzheimer’s disease (AD) and Huntington disease (HD), have a common pathological hallmark: the presence of protein aggregates. These aggregates are made of prone-to-aggregation proteins that collapse after the systems that maintain the proteome under homeostasis fail. These systems involve many cellular processes to maintain protein homeostasis under regular circumstances, and they are further induced in response to stress of many kinds (misfolded proteins, elevated temperature, etc.) including increased oxidative stress [1]. This stress is produced when there is an abnormal rise in free radicals. Free radicals are produced as a consequence of the blockage of cellular processes that impact mitochondrial function, among other processes, which in turn produces an excess of free radicals [2]. It is strongly believed that oxidative stress contributes to the progression of these diseases. Therefore, unravelling the molecular mechanisms behind oxidative stress in neurodegenerative disorders has always been considered fundamental in order to propose effective therapeutic treatment.

## 2. Oxidative Stress

Oxidative stress (OS) is defined as the imbalance of redox homeostasis due to an abnormal increase in free radicals and other reactive species, which in normal conditions play a natural role in cell signalling [3,4]. Free radicals are highly reactive species which can cause so-called oxidative damage to macromolecules like DNA, lipids and proteins.

### 2.1. Free Radicals and Reactive Molecules

Free radicals, or pro-oxidant molecules, contain one or more unpaired electrons, which is what makes them highly reactive and allows them to take electrons from other molecules [5]. They may have a different nature depending on the molecules they come from (oxygen, nitrogen, lipids, etc.). Free radicals and other reactive species are commonly generated during cellular metabolism [6]. Oxygen-derived reactive species (ROS) include free radicals and other powerfully reactive molecules such as anion radical superoxide (O_2_^•–^), hydrogen peroxide (H_2_O_2_), hydroxyl radical (^•^OH), peroxyl radical (ROO^•^) and nitric oxide (NO^•^) and are mainly produced by mitochondria. An excess of superoxide free radicals releases free Fe^2+^ from iron-containing molecules, and free iron is able to produce highly reactive radical ^•^OH through the Fenton reaction (Figure 1) (see review in [7,8]). Superoxide can react with NO to form peroxynitrite (ONOO^–^), another highly reactive and toxic free radical. Some of the ROS and nitrogen-derived reactive species (RNS), which are collectively known as RONS, can react with each other to produce other free radicals (reviewed by Weidinger and Kozlov [9]). An excess of RONS in the mitochondria produces detrimental lipid peroxidation, which increases reactive lipid species (RLS), which are a source of oxidative stress [10,11].

The source of free radicals from ROS is typically organelles with a high rate of oxygen consumption, such as mitochondria (Figure 1), endoplasmic reticulum (ER) or peroxisomes [12,13]. Free radicals can be also produced from external sources (tobacco, alcohol, drugs, pollution, fried food, etc.), although this is not the subject of this review. The best known and major source of free radicals is the mitochondria, which naturally use radicals to signal within cells. However, when the organelle is damaged (from prone-to-aggregation proteins, for example), it overproduces reactive species to toxic levels and also induces the liberation of iron from proteins (Figure 1) [2].

### 2.2. Free Radical Scavengers

The antioxidant defence system regulates free radical production to restore redox homeostasis. Natural cellular antioxidant scavengers consist of enzymatic (superoxide dismutase (SOD) [15], catalase (CAT) [16], glutathione system (GPx, GR, GST) (Figure 1) [17] and thioredoxin system (Trx) [18]) and non-enzymatic molecules. For further details on these and other enzymes dedicated to scavenging free radicals, please see [19,20]. Non-enzymatic antioxidant molecules can be exogenously provided to animal models and patients as drugs, although many are naturally acquired through the diet, like vitamins (C and E), essential fatty acids (omega-3 and omega-6), carotenoids, flavonoids and trace metals (Se, Mn, Zn) (see reviews by Ahmadinejad et al. and Halliwell [20,21]). Other antioxidants are endogenously synthesized by cell metabolism as protection against oxidative stress, like melatonin, coenzyme Q10 and reduced glutathione, among others [22,23,24,25,26,27].

## 3. Oxidative Stress and Huntington Disease

HD is a dominant inherited neurodegenerative disorder among the so-called rare diseases due to its low prevalence (affecting 1–10 of every 100,000 people in western countries). The disease is caused by an abnormal expansion of CAG repeats into exon 1 of the *HTT* gene, which encodes huntingtin (Htt), a protein whose function is still a matter of debate. Healthy people usually carry 35 or fewer CAG repeats in *HTT*, while in those who carry between 36 and 39 repeats, incomplete penetrance is shown [28]. Those who carry ≥40 CAG repeats will definitely manifest the disease with full penetrance [28]. Mutant CAG expansions encode abnormally long polyQ tracks that produce defective Htt (mutant Htt, or mHtt) with a toxic gain of function that affects the function of other non-related proteins, transcription factors and protein quality control pathways (Figure 1) [29]. There is an inverse correlation between the number of CAG repeats and age at onset of HD [30,31]. Although the specific function of Htt is still unknown, extensive research has produced data suggesting its involvement in several cellular processes, such as modulation of energy metabolism, antiapoptotic activity, transcription regulation, DNA maintenance, axonal vesicle trafficking and cell signalling [32,33,34,35,36]. Moreover, mammalian life is not possible without Htt, suggesting that its function is essential for embryonic development [37]. The aggregation process is initiated by proteolysis of misfolded mHtt monomer to produce N-terminal fragments of polyQ expanded Htt (the most toxic and worst prone-to-aggregate form) (Figure 1) [38,39]. The N-terminal fragment of mHtt contains a specific sequence (17 amino acids, N17) that acts as a nuclear import signal and favours aggregation formation in the nucleus (Figure 1), although aggregates are also found in the cytoplasm [38]. In addition, N17 encodes a specific domain that acts as an ROS sensor in order to regulate the phosphorylation and location of Htt [40]. Huntingtin is ubiquitously expressed in mammals, and this is very relevant to the design of therapeutic strategies; however, not all tissues and cells are affected equally (reviewed by Saudou and Humbert [41]).

### 3.1. Mutant Huntingtin Alters Redox Homeostasis

Medium spiny neurons (MSNs) from the striatum seem to be the cells most vulnerable to mHtt, and this sensitivity appears to depend on the extrasynaptic glutamatergic signalling sensed by N-methyl-d-aspartate receptors (NMDARs) [42]. Under normal conditions, NMDARs are activated by the release of glutamate neurotransmitter from neuronal synapses, and they induce calcium uptake to the cytoplasm, which promotes cell survival and plasticity (reviewed by Smith-Dijak et al. [43]). The expression of mHtt induces aberrant extrasynaptic glutamatergic signalling [42], which alters the NMDAR location and produces misregulation of the cAMP response element-binding (CREB)–PPARγ coactivator-1α (PGC-1α) cascade [42]. PGC-1α functions by regulating the expression of some components of antioxidant defence in mitochondria [44], which enhances overall cellular oxidative stress and contributes to cellular malfunction and eventually cell death.

mHtt expression also alters the protein quality control pathways (autophagy and ubiquitin proteasome system), transcriptional regulators and mitochondrial function [45,46,47,48]. The impairment of these processes leads to an oxidative stress condition, disrupting the redox cellular balance of proteins (Figure 1). The ubiquitin-proteasome system (UPS) and autophagy are the major mechanisms removing unfolded proteins. However, mHtt aggregates are too large to cross the proteasomal pore, so they have to be removed by macroautophagy [49]. In any case, mHtt is ubiquitinated, which affects the functionality of UPS by sequestration of molecular chaperones [46]. The autophagic marker p62 participates in the recognition of polyubiquitinated protein aggregates before autophagosome formation. It has been proposed that mHtt is associated in an aberrant manner with p62 and induces a failure of cargo recognition, subsequently producing inefficient autophagy [47]. Autophagy impairment is linked to an oxidative stress condition due to the critical role of autophagy in recycling damaged mitochondria (mitophagy) [50]. mHtt-mediated mitochondrial dysfunction is the main direct source of free radicals. Cytoplasmic mHtt aggregates associate with mitochondrial membranes, due to their lipophilic nature, and inhibit complexes II and III from the electron transport chain (ETC), which consequently reduces intracellular ATP, impairs the bioenergetic level and increases ROS (Figure 1) [51,52,53]. Increased ROS inside the mitochondria causes mitochondrial DNA (mtDNA) damage and releases free radicals to the cytosol [54,55]. mHtt destabilizes calcium homeostasis, which has a deleterious effect on mitochondrial dynamics [56]. Especially, mHtt induces a massive uptake of Ca^2+^ inside the mitochondria, mediated by NMDA receptors, which promotes an increment of mitochondrial ROS, damaging mtDNA (reviewed by Paul and Snyder [57]). Mitochondrial dynamics can be affected by physical interaction between mHtt and GTPase dynamin-related protein-1 (Drp1), increasing its expression and inducing abnormal mitochondrial fragmentation [58,59]. In contrast, mitochondrial fusion proteins such as Mitofusin 1 and 2 (Mnf 1 and 2) and optic atrophy 1 (Opa1) are downregulated in HD patients [58]. Another mHtt-related toxicity mechanism is associated with the abnormal release of metal ions out of their natural stores. Copper and iron are reactive metals that increase oxidative stress in neurodegenerative diseases such as HD [60,61].

Another source of excessive free radicals is an overload of misfolded proteins in the endoplasmic reticulum (ER) [62]. mHtt becomes cleaved by proteases (Figure 1), which liberates N-terminal fragments containing polyQs that are able to collapse into aggregates. These aggregates disrupt protein homeostasis in the cytoplasm, but also sequester endoplasmic reticulum–associated degradation (ERAD) factors [63], which induces stress on the ER. This, in turn, triggers the unfolded protein response (UPR), a highly regulated net of intracellular signals devoted to restoring protein homeostasis. The UPR consist of three signalling molecules: inositol-requiring protein-1α (IRE1α), protein kinase RNA (PKR)-like ER kinase (PERK) and activating transcription factor 6 (ATF6). Activation of the IRE1α signalling pathway regulates the fate of cell death vs. survival [64]. The activation of IRE1α results in reduced expression of nuclear factor-κB (NF-κB), which is usually activated in response to this stress [65]. A reduction in NF-κB expression leads to decreased production of the antioxidants Trx2 and Sod2, which in turn exacerbates oxidative stress and cell death [66].

Nuclear mHtt aggregates cause aberrant transcriptional dysregulation by sequestering critical transcription factors that regulate antioxidant genes (Figure 1) [67,68,69]. In addition, mHtt alters the expression of genes involved in mitochondrial biogenesis and function (Figure 1). PGC-1α is a versatile transcriptional coactivator that interacts with a wide range of transcription factors to regulate several cellular processes, such as mitochondrial biogenesis, antioxidant response and oxidative phosphorylation (OXPHOS), among others [70,71]. mHtt alters the activity of the CREB/TAF4 protein transcriptional activator complex, which regulates PGC-1α expression [69]. In addition, mHtt causes a reduction in the most potent transcriptional activator of PGC-1α, TORC1, and, inversely, induces an increment of transglutaminase (Tgase) activity, which impairs the expression of PGC-1α [72,73]. Moreover, cytoplasmic mHtt aggregates directly repress PGC-1α protein function (reviewed by Zheng et al. [74]). In both cases, the reduction in PGC-1α function causes reduced expression levels of genes that encode antioxidant enzymes such as SOD-1, SOD-2 and Gpx-1 (reviewed by Zheng et al. [74]).

As mentioned above, oxidative stress causes dramatic damage to nuclear and mitochondrial DNA [54,55,75]. Restoring this damage by the endogenous repair machinery triggers somatic expansion of CAG, since these repetitions are highly unstable (Figure 1) [76,77,78].

### 3.2. Oxidative Stress and Neuroinflammation in HD

HD patients suffer deterioration of the nervous system: some types of neurons, like medium spiny neurons, become stressed and eventually die. However, other non-neuronal types, like glial cells, are also stressed by mHtt. For example, stressed support glial cells become less efficient at removing excessive glutamate from the intercellular space, which in turn contributes to neuronal toxicity [79]. Some of these glial cells include microglia and astrocytes, the main modulator cells of inflammation in the nervous system [80,81].

Microglia, the resident immune cells of the brain, account for 5–12% of the cells in the human brain [82]. In addition to their immune status, these glial types are able to do a range of other things, like form neural circuits, maintain synapses, etc. [81]. Microglia can be found in two different forms, the so-called surveilling state and activated [81]. In the surveilling state, these cells work as chaperones for surrounding neurons, such as those described above (synapse maturation and maintenance, facilitation of neural networks by secretion of BDNF and other factors, etc.). Once they become activated, upon the detection of inflammatory stimuli, they become a main player in the neuroinflammatory response [83]. There are different kinds of inflammatory stimuli, including mHtt-induced neuronal death [83]. Once activated, they produce a range of signals such as pro-inflammatory cytokines, as well as toxic molecules like quinolinic acid and ROS to induce toxicity in invading agents [83]. They also have a phagocytic role in digesting exogenous bodies [83].

Astrocytes are among the major components of the glial population in human brains. They can be from 20% to more than 60% of total glial cells [84]. As microglia, they can be found in two states, resting and reactive astrocytes [85]. When they are not activated by a proinflammatory stimulus, they also participate in chaperone-like functions, such as synaptic formation and maintenance, and the regulation of intercellular conditions (neurotransmitter clearance, pH maintenance, ion homeostasis, etc.) [86]. They are also part of the blood–brain barrier and participate in the regulation of neurovascular function [87], among many other important roles. Once they become activated, they undergo a process called reactive astrogliosis, by which they can become harmful or protective [88,89]. Reactive astrocytes are less capable of glutamate, contributing to the toxicity of neurons, produce cytokines (pro- and anti-inflammatory) [88], and increase the amount of intercellular potassium and ROS, as do activated microglia.

It is widely accepted that neuroinflammation plays a key role in the progression of HD. Using positron emission tomography in genetically diagnosed HD patients who were pre-symptomatic, it was demonstrated that inflammation shows up before the symptoms of the disease appear [90,91]. In this regard, both microglia and astrocytes have been observed to be activated in animal models and patients with HD [83,85,88,92,93]. These cells produce harmful oxidizing agents that, together with pro-inflammatory cytokines and other toxic chemicals, induce further damage to surrounding neurons, which are already under strong internal oxidative stress induced by mHtt. At the same time, the oxidative stress on these toxified neurons also signal the activation of microglia and astrocytes [94], which undergo the changes described above to produce and secrete toxic molecules (pro-inflammatory cytokines, ROS, etc.). In parallel, a high concentration of reactive species produced by neurons and immune cells activates signalling pathways that maintain the secretion of proinflammatory cytokines and chemokines [95], further contributing to the vicious cycle and making inflammation and toxicity chronic.

### 3.3. Oxidative Stress in Huntington Disease Patients

From the 1990s, when the causative gene of HD was discovered, to date, many experiments have been conducted to find out whether HD patients suffer some sort of oxidative stress. Although early attempts to find biological markers of oxidative stress failed, the signs of such stress showed up when technology and methods of detection become more sensitive. For example, Browne and co-workers showed that postmortem HD brains contained abnormally high amounts of biomarkers indicating that they had gone through oxidative stress, like higher expression of heme oxygenase (an oxidative stress enzymatic scavenger) or 3-nitrotyrosine (a marker of nitrogen-derived free radicals) [96], as happens in patients with ALS, a disease whose progression is believed to be fuelled by oxidative stress [97]. Sorolla and co-workers also found robust data showing that the striatum regions of the brains of HD patients are under oxidative stress, since they discovered strong activation of antioxidative stress enzymes (peroxiredoxins, glutathione peroxidases, SOD, etc.) [98]. Those authors also found carbonylated proteins, a well-known sign of oxidative stress [98]. The free radicals which may induce such stress are potentially produced by mitochondria (Figure 1). In this regard, Polidori et al. found evidence of mitochondrial oxidative stress (8-hydroxy-2-deoxyguanosine in mtDNA) in the cortex of HD brains [54]. The same biomarker was confirmed to be elevated in the caudate nucleus in post-mortem brains of HD patients [99] and circulating blood [100]. Although these findings have been challenged by other authors [101], later research validated some of them [102,103]. In fact, Duran and co-workers validated the occurrence of some of these circulating biomarkers and described a few more, strongly suggesting that circulating blood carries plenty of signs of oxidative stress [103]. Free radicals and elevated oxidative stress scavengers have also been shown in mitochondria of cultured fibroblasts from patients [104]. Other authors found that mHtt localized in sites of DNA damaged by oxidative stress, in fibroblasts from HD patients, and that oxidative stress was a driver of the progression of HD [78].

As shown above, there is consensus about the presence of free radicals and oxidative stress in different tissues of HD patients, but where do the free radicals and other reactive species come from? One sure source is mitochondria. Both animal models (see the following section) and patients with HD show signs of defective mitochondrial function. Lymphoblasts from HD patients contain mitochondria with reduced enzymatic activity in key components of the Krebs cycle, which in turn disrupts basal respiration [105].

Metal homeostasis imbalance has been observed in brains of HD patients, and since they are catalysts that induce the production of free radicals, they are believed to participate in the progression of the disease [106]. In this regard, iron and copper accumulation has been observed in HD patients, both in post-mortem tissue [61] and using magnetic resonance imaging [107]. Both metals can catalyse the production of free radicals by the Fenton reaction (free iron; Figure 1) [8] or through a Fenton-like reaction (copper) [108].

Taken together, the data strongly suggest that oxidative stress happens in people suffering from HD, and that may play a key role in the progression of the disease. Therefore, many researchers followed the next logical step, which is to perform experiments with animal models of HD, to further explore the presence of free radicals, find out potential mechanisms by which oxidative stress participates in the disease, and test antioxidant therapies to reduce this stress.

### 3.4. Oxidative Stress in Animal Models of polyQ Toxicity and Huntington Disease

#### 3.4.1. Using *Caenorhabditis elegans* Models of polyQ Toxicity to Investigate Antioxidants as a Therapeutic Intervention

*Caenorhabditis elegans* is a microscopic round nematode that was established by Sydney Brenner in the 1970s as a model organism to study animal development and the function of the nervous system [109]. Later, as its sequenced genome became available, it was obvious that it would be very useful to study human diseases, as it is estimated that 42% of human genes that cause diseases have an orthologue in *C. elegans* [110]. Many worm models of polyQ disorders recapitulate phenotypes observed in diseases such as HD and some spinocerebellar ataxias (SCAs), among other disorders (reviewed by Rudich and Lamitina [111]). The readouts in polyQ models when assaying drugs or genetic modifiers depend on which tissue the polyQs are expressed on. For example, worms that express polyQs fused to fluorescent proteins in muscle cells allow for investigation of the dynamics of polyQ aggregation and motor function. In contrast, when polyQs are expressed in neurons, complex behaviour such as mechanosensation or chemosensation can be studied [111]. Following this logic, many compounds have been assayed in different worm models of HD that express a track of polyQ in muscle cells (AM141) and ASH sensory neurons (HA759). AM141 contains a transgene that expresses 40 glutamines (40Q) in a frame with a fluorescent protein (YFP) in muscle cells driven by the promoter of *unc-54* (*unc-54p::40Q::YFP*) and shows an age-dependent aggregation pattern [112]. HA759 animals (*osm-10p::GFP* + *osm-10p::Htn150Q* + *dpy-20*(+); *pqe-1(rt13)*) encode the N-terminus of human Htt carrying a long polyQ tract (Htn150Q) fused to GFP and are expressed mainly in the polymodal sensory ASH neurons (promoter of the gene *osm-10*) [113] (Table 1). This strain also carries a mutation in the *pqe-1* gene that sensitizes the ASH neurons to cell death, and it is a suitable model to measure neuronal dysfunction [114].

These models show signs of oxidative stress, since ablation of the systems to buffer free radicals enhances the phenotypes caused by polyQ aggregation and other prone-to-aggregation peptides [124]. Machiela and co-workers found out that the expression of 40Q in muscle cells induced increased sensitivity to oxidative stress [115]. These animals presented an increase in ROS levels and upregulation of SOD and CAT. However, when *sod* genes were ablated, the animals did not have enhanced aggregation. The treatment of these worms with three antioxidants (vitamin C, α-lipoic acid and epigallocatechin gallate (EGCG)) also failed to ameliorate movement impairment or aggregation of polyQs [115]. Aside from being a potent antioxidant, EGCG is also an activator of AMP-activated protein kinase (AMPK) [125,126], a master regulator of energy and metabolism. In contrast with the Machiela study, the activation of AMPK using metformin has been shown to improve aggregation of polyQs in worms expressing 40Q in muscle cells [127]. However, this is not always the case. Singh et al. assayed phycocyanin, an antioxidant protein from phycobiliproteins plants [128], in worms expressing 40Q in muscle cells (AM141) [120]. In contrast with the study by Machiela et al., phycocyanin significantly reduced the number of inclusion bodies in these animals [120]. These apparent conflicting results, and the results reported for EGCG, show the complexity of using drugs that may be pleiotropic. One antioxidant may not have an effect because of its antioxidant power, but because it activates a signalling pathway that leads to cell protection (e.g., autophagy).

Most of the strategies to fight oxidative stress in worms were based on using substances or extracts from plants, for example, salidroside, a phenol glycoside obtained from *Rhodiola rosea*, a medicinal plant traditionally used in China as an anti-fatigue herb and in the West to treat anxiety and depression. Its antioxidant potential has been validated in vitro in fibroblasts from HD patients [129] and in vivo [121]. Salidroside decreased lipid peroxidation and ROS, increased antioxidant enzyme activity and neuronal survival, and alleviated polyQ-induced behaviour deficiency in a polyQ neuronal model (*osm-10p::**Htn150Q*), but it was not able to prevent polyQ aggregation in muscle cells [121]. Polysaccharides from *Dictyophora indusiata*, a mushroom, have been traditionally used as a medicine to treat inflammatory and neural diseases [130,131]. The polysaccharides exhibited antioxidant capacity in vitro and in vivo [122,132]. *Dictyophora indusiata* polysaccharides were found to attenuate polyQ-mediated chemosensory dysfunction in *C. elegans* ASH neurons [122]. Another plant acidic polysaccharide from *Epimedium brevicornum*, EbPS-A1, a traditional Chinese medicine, is able to scavenge free radicals, producing a decrease in ROS levels and lipid peroxidation. In animals expressing polyQs in ASH neurons, EbPS-A1 increased the survival rate under stress-induced conditions, reduced ROS levels and lipid peroxidation, and increased SOD and CAT. While it did not reduce polyQ aggregates in muscle cells, it alleviated polyQ-mediated chemosensory dysfunction in ASH neurons [117], as also happened with salidroside. The well-known guarana extract is mainly composed of caffeine, methyl-xanthines and poly-phenols, which have an effect on biological processes including oxidative stress [133]. Culturing worms on this extract alleviated polyQ-mediated chemosensory dysfunction in ASH and reduced polyQ aggregation in muscle cells [118]. Extracts from *Cassia fistula*, a tree from tropical Asia, contain mainly phenolic compounds with powerful antioxidant properties [134]. Treating worms with the extract reduced the number of polyQ inclusion bodies in muscle cells [116].

Aside from plant extracts, synthetic compounds like ruthenium (II) complexes were used to treat worms. These substances regulate stress response pathways through the JNK-1/DAF-16 signalling pathway [119]. Treating worms with this substance reduced polyQ aggregation in muscle cells, but showed less polyQ-mediated death of ASH neurons and improved chemosensory behaviour [119].

#### 3.4.2. Studies Using Antioxidants to Treat *Drosophila melanogaster* Models of HD

*Drosophila melanogaster*, like *C. elegans*, is very useful model organism to investigate the pathological mechanisms behind HD- and polyQ-related disorders (Table 2) [135]. Two HD models of flies have been used to study oxidative stress and treatments for it: Htt-128Q and Httex1p 93Q (Table 2). Htt-128Q flies express the N-terminal 548 amino acids of the *Htt* gene with 128Q, under the control of a heat shock promoter. This fragment from mHtt is considered to be particularly pathogenic in many animals and in humans [136]. In flies, this molecule induces premature death, deficits in behaviour and eye colour or uncoordinated movement [137]. The Httex1p 93Q fly model expresses exon 1 of Htt together with 93Q in photoreceptors [138], which is also considered very toxic. These flies show loss of photoreceptors, neural degeneration and reduced mobility and lifespan [138]. There are no data regarding the two fly models described above showing evidence of oxidative stress. However, oxidative stress has been observed in other *Drosophila* strains expressing polyQs in cardiomyocytes, suggesting that THE expression of polyQs induces oxidative stress in flies as well [139]. In those flies, higher ROS levels and mitochondrial dysfunction markers were detected, and these effects were reversed by overexpression of SOD antioxidant genes [139]. To explore potential antioxidant treatments, α-tocopherol (vitamin E) and coenzyme Q10 (a key player in the electron transport chain of mitochondria) [140] were assayed together in the Htt-128Q model. However, these antioxidant substances did not increase survival rates and pupal mortality, suggesting that they cannot reduce the toxicity of this highly deleterious mHtt fragment [141]. However, treating the Httex1p 93Q flies with fisetin and resveratrol produced neuroprotective and neurotrophic effects through activation of the ERK pathway, which resulted in better cognition [142,143,144]. These substances are polyphenols found in fruits that reduce oxidative stress by scavenging free radicals and reducing lipid peroxidation [145]. In other experiments, fisetin alone was able to induce a dose-dependent increase in photoreceptor neuron survival and overall survival of animals in the same HD model [144], and treating them with just resveratrol was able to rescue neuronal degeneration in a dose-dependent manner [146]. Grape seed polyphenolic extract (GSPE) is a strong antioxidant and powerful metal chelator that extended longevity in Httex1p 93Q flies compared to controls [147]. Curcumin, a polyphenolic compound from turmeric, has antioxidant properties which can reduce nitric-oxide-derived free radicals (RNS) [148] and other reactive species [149]. The administration of curcumin to Httex1p 93Q animals showed a decrease in photoreceptor loss and improved polyQ-induced motor neuronal dysfunction [150]. Green tea is an excellent source of polyphenol antioxidants, such as catechins, and its administration upregulated SOD and CAT expression in *D. melanogaster* [151]. Recently, Varga and co-workers used green tea supplementation in Httex1p 93Q flies, reducing polyQ-induced neurodegeneration and increasing lifespan [152]. Treatment with one of the main polyphenolic compounds of green tea, epigallocatechin-3-gallate (EGCG), which failed to improve polyQ-related phenotypes in *C. elegans* [115], inhibited the aggregation of Httex1p-93Q in a dose-dependent manner and improved photoreceptor degeneration and motor function [153]. These results were in agreement with previous results by Sanchis and co-workers showing that AMPK activation by metformin reduces polyQ aggregation in *C. elegans* [127].

#### 3.4.3. Oxidative Stress and Antioxidant Therapies in Rodent Models of HD

Rodent models of HD are frequently used to test drug and other interventions, because they are one of the easiest to handle mammalian models. Moreover, they show similar phenotypes that somehow correlate with the traits observed in HD patients [154]. Some of these models show altered expression of NOS [155] and SOD [156] and increased the production of ROS [157] and lipid peroxidation [158]. This is very relevant, because it allows preclinical studies using antioxidants to treat oxidative stress induced by mHtt, and is essential to set the basis for future clinical trials in HD patients. For simplicity, in this review we focus on studies using antioxidants performed relatively recently. All of these interventions are described in Table 3. Two types of rodent models were used: genetically modified (R6/1 mice, R6/2 mice, N171-82Q mice, YAC128 mice, *Hdh*^(CAG)150/(CAG)150^ mice and CAG140 knock-in mice) and chemically induced (3-nitropropionic acid (3-NP) rats, malonic acid- and quinolinic acid-induced rats).

R6/1 mice carry an insertion of a transgene which drives the expression of exon 1 of the human *HTT* gene, containing 115 CAG repeats [194]. These mice show progressive oxidative striatal damage, as judged by the presence of lipid peroxidation in this brain region [195]. R6/2 mice are similar to R2/1, since they carry an insertion of exogenous DNA which encodes exon 1 of the human *HTT* gene, with a longer repeat expansion (145 CAG) [194]. This mouse model recapitulates many HD symptoms [194] and shows a substantial amount of ROS in neurons [169]. N171-82Q mice express the N-terminal fragment of human *HTT* containing 171 amino acids with 82Q. These mice show motor and behavioural deficits and have increased malondialdehyde (a biomarker of lipid peroxidation) and 8-hydroxy-2-deoxyguanosine (OH^8^dG) as a result of DNA oxidation damage [196,197,198]. In contrast with previous models, YAC128 mice carry an insertion with a yeast artificial chromosome containing a full-length *HTT* with 128Q [199]. These mice show hyperactivity in early stages, followed by the onset of motor problems and finally hypokinesis [200]. These mice were assessed for oxidative stress biomarkers, but their levels of lipid peroxidation and protein carbonyl formation were similar to their wild-type littermates [201]. *Hdh*^(CAG)150/(CAG)150^ mice contain an abnormal expansion of 150 CAG triplets into murine huntingtin gene, *Hdh* (homologous human *HTT* gene), which induces an increase in oxidative stress that produces mitochondrial DNA damage [202]. These mice also showed decreased aconitase 2 activity within the striatum in 16-month-old animals, which is an indirect indication of oxidative stress [203]. The last model, CAG140 knock-in mice, express a chimeric mouse/human exon 1 with 140 CAG repeats inserted into the murine huntingtin gene, *Hdh*. CAG140 knock-in mice show pathological, molecular and behavioural deficits similar to what happens in patients with HD [204]. These mice also show elevated levels of OH^8^dG in urine and brain tissue [205].

Regarding the chemically induced models, the 3-nitropropionic acid (3-NP)-induced HD rat model is characterised by chorea and cognitive deficits produced by the administration of this toxic substance, which acts by inhibiting succinate dehydrogenase of the Krebs cycle [206,207]. 3-NP causes mitochondrial dysfunction, which can be seen as an indirect biomarker of the presence of oxidative stress, among other consequences (reviewed by Damiano et al. [208]). Another chemically induced model is produced by injections of quinolinic acid, which acts as an NMDAR agonist, producing an imbalance in calcium homeostasis [209] that produces DNA mitochondria damage and neuronal loss in rats [210,211,212]. Finally, some authors used malonic acid-induced rats [191]. This acid, injected within the striatum, induces loss of body weight, disrupts motor coordination and causes oxidative stress. These rats show higher oxidized glutathione, malondialdehyde and nitrite levels, together with reduced superoxide dismutase and catalase function [191].

Many strategies to treat rodent models of HD in order to reduce oxidative stress were based on natural metabolites and hormones. Vamos and co-workers analysed the effect of l-carnitine on N171-82Q mice. l-carnitine is a modulator of lipid metabolism [213] and has been shown to reduce oxidative stress by scavenging free radicals [214]. The administration of this substance increased survival rate, reduced the number of aggregates of N171-82Q and improved motor performance [173]. Treating 3-NP-induced rats with N-acetylcysteine (NAC), a precursor of cysteine and a powerful antioxidant, prevented mitochondrial dysfunction, neuronal death and attenuated lipid peroxidation, among other things [184]. NAC reversed depressed-like behaviours in R6/1 mice, decreased oxidative damage markers in mitochondria and delayed the onset of motor defects [159,160]. Melatonin is a hormone produced by the retina and the pineal gland [215], is mainly found in the brain [216] and has high antioxidant capacity [217]. Treatment with melatonin in R6/2 produced a decreased mortality rate and delayed onset of the disease [170].

Other strategies involved the use of natural cofactors like α-lipoic acid, which is as a cofactor for pyruvate dehydrogenase and α-ketoglutarate dehydrogenase in mitochondria. This chemical is also a quencher of ROS [218] and a metal chelator [219]. Treatment with α-lipoic acid decreased mitochondrial swelling produced by 3-NP and reduced cognitive impairment in 3-NP-induced rats [180]. 3-NP-induced rats were also treated with nicotinamide, a form of vitamin B3, and the authors observed that it prevented an increase in nitrite levels and malondialdehyde and induced a depletion of GSH, which indicates reduced pressure of oxidative stress. Moreover, nicotinamide improved motor performance and impeded neuronal death in the striatum [185].

As we described with regard to *C. elegans* and *Drosophila*, plant extracts were also used to treat HD mice and rats. Lycopene, a major carotenoid found in tomato, is responsible for its colour and considered a powerful antioxidant [220]. In 3-NP-induced rats, lycopene was able to ameliorate mitochondrial function and prevent locomotor and memory deficits [183]. Grape seed polyphenolic extract was orally administered to R6/2 mice, producing an increase in lifespan and better performance on motor coordination tests [147]. YAC128 mice were also successfully treated with resveratrol, which improved their motor performance and learning abilities [176]. Quercetin is the most potent scavenger of RONS of the flavonoid family. Treating 3-NP-induced rats with this substance reversed mitochondrial dysfunction, prevented a decrease in antioxidant enzyme activity and improved motor performance [186]. Moreover, a combination of quercetin and lycopene alleviated anxiety and depression in 3-NP-treated rats [187]. Sulforaphane, an isothiocyanate found in cruciferous vegetables, was administered to quinolinic acid-induced rats, preventing mitochondrial dysfunction [193]. Rutin, a flavonoid obtained from buckwheat and a free radical scavenger, was used to treat 3-NP-induced rats. Rutin increases the production of GSH, SOD and CAT and reduces xanthine oxidase function, which participates in ROS generation [221,222,223]. Treatment of 3-NP-induced rats with rutin protected these animals against body weight loss, restored the activity of antioxidant enzymes and improved motor and memory performance [188]. Treating CAG140 knock-in mice with polyphenolic curcumin diminished the number of Htt aggregates in the striatum, but the authors only observed a partial improvement in motor function [179]. Recently, Elifani and co-workers described that R6/2 mice, when fed with curcumin from birth, showed better motor performance and activation of pro-survival pathways such as ERK. These mice also presented an imbalance in gastro-intestinal homeostasis, which causes body weight loss when it is disrupted. Curcumin treatment of R6/2 mice maintained gastro-intestinal homeostasis [165], suggesting that further investigation may lead to useful outcomes.

Other strategies to fight oxidative stress in rodent models of HD consisted of using drugs of synthetic origin. Treatment R6/2 with cystamine, an organic disulfide, was able to extend lifespan and reduce motor deficits [166] by producing increased levels of cysteine, which is responsible for the antioxidant effects [167]. FK-506, or tacrolimus, is an immunosuppressant used to prevent allograft rejection. It prevented body weight loss, rescued motor activity, restored antioxidant enzyme activity and attenuated mitochondrial dysfunction in 3-NP-treated rats [182]. Dimethylfumarate (DMF) is a fumaric acid ester that activates the NF-E2 related factor 2 (Nrf2) antioxidant pathway [224]. This pathway is relevant in many neurodegenerative disorders, and it has been shown to be impaired in models and patients with HD (reviewed by Liddell [225]), hence inducing the pathway may be protective. In agreement with this prediction, treatment with DMF induced extended lifespan in R6/2 mice and beneficial effects on the motor behaviour and preservation of neurons in R6/2 and YAC128 mice [169]. However, YAC128 mice up to 12 months were assessed for oxidative stress, and it was found that lipid peroxidation and protein carbonyl formation in different brain sections remained the same compared to wild-type mice [201]. Therefore, it is essential to further investigate how DMF induces beneficial effects in these animals, because it might be activating other pro-survival pathways not related to oxidative stress. Deferoxamine, a potent iron chelator, was used to treat R6/2 mice after iron accumulation was discovered [168]. Although there were promising results regarding behavioural performance, deferoxamine does not pass the blood–brain barrier and has not been further studied as an alternative treatment for HD [168]. 3-NP-treated rats were also treated with bis selenide, a selenium derivative with antioxidant properties, which attenuated body weight loss, induced better motor performance and restored CAT enzymatic levels [189]. Moreover, the administration of sodium selenite, an inorganic form of selenium, to N171-82Q mice improved motor behaviour, reduced N171-82Q aggregates and decreased oxidized glutathione levels in the brain [174]. XJB-5-131 is a synthetic antioxidant conjugated with a mitochondria-targeting moiety. When this substance was administered to R6/2 and *Hdh*^(CAG)150/(CAG)150^ mice, it prevented mitochondrial dysfunction and oxidative damage in the latter and prevented weight loss and improved motor performance in both strains of mice [171,177,178].

Although these experiments prove that oxidative stress affects murine models of HD and that treatment with substances that have antioxidant properties can restore functionality to the animals, they were not taken to the bedside to investigate their effect on HD patients. However, other experiments and treatments with rodents were taken to clinical trials. For example, α-tocopherol (vitamin E) combined with coenzyme Q10, which are well-known potent antioxidants and key coenzymes of metabolism, was first administered to 3-NP-induced rats [181]. In particular, coenzyme Q10 is vital for the appropriate transfer of electrons in the ETC in the mitochondria, so the authors expected a restoration of the 3-NP-induced decline in mitochondrial production of energy after treating these rats with 250 mg CoQ10 + 530 mg vitamin E/kg/day. However, they did not observe a substantial rescue of energy production in these organelles [181]. Another laboratory assayed α-tocopherol in malonic acid-induced rats [191]. Malonic acid, injected within the striatum, induces loss of body weight, disrupts motor coordination and causes oxidative stress. However, treatment of these rats with α-tocopherol at 50 or 100 mg/kg/day reversed these processes by preserving mitochondrial function [191]. Other authors assayed the addition of 0.2% mitochondrial cofactor coenzyme Q10 in the diet in the R6/2 mouse model of HD [163]. This treatment increased survival of the mice and delayed the appearance motor deficits, reduced weight loss and cerebral atrophy, and there were fewer intranuclear inclusions of mHtt [163]. Creatine, in addition to being an antioxidant, is believed to fuel mitochondrial oxidative phosphorylation, hence raising the energy level in cells. This substance was assayed in two mice models of HD, and both showed that it was neuroprotective at the cellular and organismal levels. In the first study, Ferrante et al. showed that creatine was able to reduce aggregates of mHtt, which in turn improved the survival of the animals and delayed mHtt-induced atrophy of striatal neurons of R6/2 mice [164]. In another study, the same laboratory showed that creatine induced similar positive effects in N171-82Q mice: improved survival rate, delayed motor dysfunction and weight loss, with reduced aggregated mHtt [172]. Idebenone is a synthetic analogue of coenzyme Q10, since it contains the quinone moiety responsible for its antioxidant capacity, but it differs in the length of the lipophilic arm [226]. This gives this substance many different physicochemical properties than coenzyme Q10, for example, higher solubility in aqueous solutions, and therefore different subcellular localization (coenzyme Q10 is mostly mitochondrial, while idebenone is widespread) [226]. Idebenone was used as an antioxidant to treat rats with brain lesions induced by three excitatory amino acids (quisqualate, kainate and quinolonate) [192]. This treatment was neuroprotective against quisqualate- and kainite-induced lesions, while it did not protect against quinolonate [192]. Polyunsaturated fatty acids are well-known antioxidant agents. Based on this, a preclinical trial was performed using virgin olive oil to treat 3NP-induced rats, and this product was shown to be a powerful brain antioxidant [190]. Other preclinical assays were done in genetic R6/1, R6/2 and YAC128 mouse models. Using α-lipoic acid as an antioxidant, Andreaassen et al. observed a modest improvement in the survival of R6/2 mice [162]. Other authors found higher survival rates and better motor performance in R6/1 mice using 48% linoleic acid, 6% Q-linolenic acid, 3% eicosapentaenoic acid, 2% docosahexaenoic acid, 5% K-lipoic acid and 3% d-K-tocopherol as treatment [161]. Finally, some authors treated YAC128 mice with 1% ethyl-eicosapentaenoic acid, which resulted in a modest improvement of motor function [175].

Altogether, these data show sometimes controversial, yet hopeful, results, representing the basis of future clinical trials using some of these substances as therapeutic interventions to treat HD in patients.

### 3.5. Clinical Trials Using Antioxidants as a Therapeutic Intervention to Treat HD

As shown above, there are convincing data showing that oxidative stress participates in the progression of the disease in animal models and patients with HD. All of these data, and many more not necessarily related to oxidative stress events, provided the rationale for a vast number of clinical assays of different nature. Since it is not the purpose here to provide an exhaustive review of all studies conducted to date, we focus on assays targeting oxidative stress. The clinical trials of HD using antioxidants described below are detailed in Table 4.

The promising results with α-tocopherol obtained in malonic acid-induced rats (see previous section) [191] were the basis on which to perform a clinical study. In the study, 40 HD patients were treated with 3000 IU vitamin E/daily and 33 patients were treated with sham. The assay demonstrated that there were no significant effects overall between the groups on neurologic and neuropsychiatric symptoms. However, the authors observed a therapeutic effect on neurologic symptoms for patients in the early phases of HD [227].

The positive results obtained with coenzyme Q10 on R6/2 mice (see above) [163] encouraged the authors to perform a clinical trial using this coenzyme. Firstly, they assayed several doses in a phase I trial (600 mg, 1200 mg and 2400 mg daily) with 45 HD patients, while 45 patients constituted the control arm (PREQUEL trial, Clinicaltrials.gov, number NCT00920699). The drug was well tolerated, although no signs of functional improvement were found [228]. Therefore, they escalated the trial to 240 treated HD patients and 240 controls, in the 2CARE assay (Clinicaltrials.gov, number NCT00608881) [229]. An analysis of the results failed to show any benefit in patients with HD treated with 2400 mg coenzyme Q10/day [229].

Several studies have been done using creatine as an antioxidant drug to treat HD patients [100,230,231,232]. Some were small studies that, after some apparent success, fuelled bigger and more ambitious assays. As with the substances described above, the motivation behind these studies was data from experiments performed in mice models of HD (see previous section). The first study was done using 8 g of creatine daily for 16 weeks [100]. They found that this dose was safe and well tolerated, and they observed a reduction in OH^8^dG, an oxidative stress biomarker in circulating blood [100]. This was followed by an open-label study with some of these patients with 30 g daily, showing that the higher dose was well tolerated [231]. This dose was also able to slow down cortical atrophy [231]. In light of these positive results, more ambitious studies were planned: the PRECREST trial (Clinicaltrials.gov, number NCT00592995), with 47 HD patients on creatine and 17 HD untreated controls, which tested larger doses [232], and the CREST-E trial (Clinicaltrials.gov, number NCT00712426) with many more patients (275 treated HD patients and 278 untreated) [232]. Despite these efforts, no differences between groups were seen, and this substance was abandoned as a therapy to treat HD.

Following the results obtained in murine models of HD (see above), Ranen et al. performed a clinical trial using idebenone to treat patients with HD. They administered 90 mg three times a day to 50 patients, while 50 HD patients were given placebo [233]. Even though the basic science behind it was good and the rationale to do the clinical study strong, the result was deceptive since no differences were found between groups.

Based on work in mice (previous section), some clinical trials have been performed to investigate whether polyunsaturated fatty acids may be of use to treat HD. The first assay was done by Vaddadi and co-workers, treating 10 patients (7 controls) with 8 g of HUFAs daily (70 mg γ-linolenic acid, 35 mg eicosapentaenoic acid and 20 mg docosahexaenoic acid, with 50 mg α-lipoic acid and 30 mg vitamin E, and the remaining lipids as carriers) [234]. This treatment produced an improvement in motor behaviour [234]. Another small study using 8 g of ethyl eicosapentaenoate to treat seven HD patients (four controls) also showed motor improvement as well as MRI changes in the brain [235]. These authors pursued a bigger assay, recruiting 67 HD patients (with 68 patients in the control arm) for treatment with 2 g of ethyl eicosapentaenoate, which further supported the motor improvement of the previous study [236]. Finally, a much bigger assay was done, TREND-HD, using 1 g of the same substance to treat 150 HD patients (150 controls) (Clinicaltrials.gov, number NCT00146211) [237]. Unfortunately, this clinical trial showed no differences between groups [237].

OPC-14117, a synthetic free radical scavenger that tends to accumulate in the brain, was used to treat 32 HD patients (32 controls) [238]. The authors used 60, 120 and 240 mg a day, and although the compound was well tolerated, no significant differences between groups were found. 

Altogether, this adds up to somewhat poor results, since none of these substances are currently used to treat HD, so new, better antioxidants will need to be developed. Alternatively, other strategies may be considered in the pursuit of finding a cure for HD. This is discussed in the last section of this review.

## 4. Conclusions and Remarks

### 4.1. The Antioxidant Approach to Treat HD Is Promising but Requires Further Refinement

The volume of data regarding the involvement of oxidative stress in HD is overwhelming. This kind of stress has been extensively demonstrated in both animal models and patients with HD. Moreover, treating animal models of HD with compounds that reduce oxidative stress has been shown to reduce free radicals and other reactive species, but more importantly, these treatments were able to restore some sort of functionality, from motor improvement to cognitive recovery. Therefore, the foundations on which the clinical trials were designed were solid and indisputable. Some of these substances, like α-tocopherol, managed to reduce some phenotypes in HD patients in the early phases of the disease. However, many intelligently designed clinical trials, testing anti-oxidative stress strategies, did not produce substantial motor, cognitive or psychiatric functional recovery of HD patients, although some drugs succeeded in alleviating symptoms [240]. In some cases, signs of oxidative stress were reduced, but no impact on disease progression was observed [101,228]. In other cases, moderate signs in secondary targets showed that the treatment may have been slightly beneficial at only modest levels, only to point out that better approaches were required to make these approaches useful [231]. Why is this? There may be a number of reasons.

On one side, animal models may not accurately reproduce the disease, so the findings do not translate well to humans. Moreover, drug doses that work in mice, for example, reducing free radicals in the brain, may not be translatable to humans due to the obvious differences in size and weight. The means to deliver the drugs may also not be feasible in humans, such as repeated intraperitoneal injections. Finally, the metabolic rate of rodents is faster than ours and they process therapeutic substances differently. Aside from these interspecies differences, other, deeper causes may be responsible for the unsuccessful attempts to cure HD by targeting oxidative stress. In the brains of HD animal models, free radicals and reactive species appear in the sets of neurons affected by mHtt. Therefore, when administering an antioxidant that can be taken orally and travel across the blood–brain barrier, it will target all neurons, not only the ones affected. This has different implications. For example, the antioxidant may not reach the target neurons in the right concentration, since many non-affected neurons surrounding the sick neuron are also exposed to the drug. Secondly, some glial cells may have defective signalling through free radicals, which are buffered by antioxidants and hence do not properly work to keep the surrounding neurons healthy (see review about the importance of glial cells to neighbour neurons [241]). Moreover, free radicals, naturally produced by humans during exercise, are promoters of the organism’s health, because they induce the expression of protective genes (antioxidants, chaperones, etc.), and using exogenous antioxidants during exercise negates the benefit of this activity [242]. Furthermore, oxidative stress does not happen homogeneously through the entire cell; it is restricted to organelles (mitochondria, etc.) and neighbouring regions, but some drugs may not be able to target these places.

As we have discussed, it is not an easy task to make substantial changes in the progression of HD just by using antioxidants as therapeutic agents. Given these somewhat discouraging circumstances, what can be done to properly address finding a treatment to cure HD? From the point of view of fighting oxidative stress, one sure thing is to get better antioxidants. In this regard, many elegant approaches have been designed to refine the efficiency of antioxidants. The development of a new generation of mitochondria-targeted antioxidants raised the hope of solving this issue. For example, MitoQ (mitoquinone) is a ubiquinone-derivative chemical that is targeted to the mitochondria by means of the lipophilic triphenylphosphonium cation, which carries attached covalence [243]. As a ubiquinone, this molecule has a strong antioxidant power, but it concentrates mostly in the mitochondria, thereby reducing the reactive species within this organelle [243]. This has been shown to work in in vitro models of HD [244]. Another example is the mitochondria-targeted antioxidant SkQT1, which can cross the blood–brain barrier in rats, and reduces phenotypes in a model of amyloid-beta1-42 (Abeta)-induced impairment of long-term hippocampal potentiation [245]. This compound is a derivative of thymoquinone, a potent antioxidant quinone from the seed of black caraway (*Nigella sativa*) [246]. There are more of this kind, and this subject has been reviewed by Oyewole and Birch-Machin [247]. Even though these authors took a clever approach to reducing oxidative stress, very modest outcomes have been seen in neurodegenerative diseases using these substances in in vitro and in vivo studies. Therefore, these approaches will require further work to be useful as therapies.

### 4.2. Do We Really Want to Remove Free Radicals, or Something Else?

Targeting oxidative stress lessens just one of the many toxic effects induced by mHtt. This mutant protein disrupts many cellular processes: transcriptional dysregulation, defective axonal transport, protein homeostasis impairment and mitochondrial malfunction, which in turn induces oxidative stress, drops in energy levels and pro-apoptotic signals, and more. All of these events are intrinsically related, so there is intense cross-talk between them. Therefore, one antioxidant drug may be able to temporarily reduce the presence of free radicals, but since many other sources of stress remain and they may be further fuelling the production of reactive species, these medicines may be incapable of maintaining a sustained reduction in the toxic species. One way to address this problem is by targeting the source of all cellular toxic issues, which is mHtt (Figure 2). This can be done by inhibiting the production of mHtt, and different approaches may be useful, depending on the cut point to stop the expression of huntingtin.

One way to stop the expression of mHtt would be to use genetic engineering with clustered regularly interspaced short palindromic repeats (CRISPR) to edit the mutant allele and remove the abnormal CAG expansion (Figure 2). Alternatively, CRISPR could be used to induce a knockout of the gene, preferably to the mutant allele, so the wild-type allele would still be expressed. CRISPR is a prokaryotic immune system that confers resistance to invasion from plasmids and phages (reviewed by Jiang [248]). The components of this system includes RNA fragments, which can guide a DNA nuclease (Cas9, for example) to any 20 bp of DNA in a given genome so the nuclease can induce a double-stranded cut [248]. Once the cut is done, the endogenous machinery of DNA repair will use the sister chromatid as a template to copy the sequence and repair it, or a non-homologous end joining repair will introduce small deletions or insertions (virtual knockout) [248]. This is under heavy research in many labs (see, for example, [249,250]), and it is a promising approach since it would represent a definitive cure for HD, but the technique is in its infancy. Therefore, many refinements need to be done to avoid undesirable consequences of its use, such as causing off-target mutations. Other issues also remain, such as how to deliver the CRISPR components within the right cells in the brain.

Other ways to silence mHtt are more affordable, such as the use of RNAi or single-stranded oligonucleotides (ssODNs) that target the nascent RNA encoding huntingtin (Figure 2). This can also be done in a mutant allele fashion [251], given that some SNPs are present during heterozygosis in the *HTT* sequence. Much research has been done using them on animal models of HD [252]. These techniques are much more mature than CRISPR, and this is why ssODNs are under investigation as therapy for HD in clinical trials (ClinicalTrials.gov, number NCT02519036) [253]. This is a lowering-all-huntingtin study that does not discriminate between alleles [253]. However, there is ongoing work in performing mutant allele-specific huntingtin repression studies using antisense oligonucleotides (see, for example, ClinicalTrials.gov, number NCT03225846) (some revisions on the topic [254,255]).

Next, targeting the protein is also feasible, although very challenging. In this regard, antibodies fused to ubiquitin ligases can be engineered to target toxic proteins and send them to the proteasome (Figure 2) [256]. They could be designed to mark mHtt specifically and send it to degradation. However, as mentioned above, this needs heavy research to be of any use as a therapy. Another, much easier way is the use of drugs to actively influence the degradation of mHtt. This can be accomplished using substances that induce activation of autophagy (Figure 2), a natural process of the cell by which mHtt is eliminated. Moreover, in animal models and patients with HD, this process is inhibited by mHtt itself [257]. However, some authors have demonstrated that AMPK activators (Figure 2) are able to reduce the amount of mHtt in different models of HD [127,258,259]. Moreover, metformin, a well-known activator of AMPK and a worldwide treatment against type 2 diabetes, is able to delay cognitive decline in patients with HD (Figure 2) [260]. Finally, therapies using ssODN target mHtt exclusively in the nervous system, but huntingtin is a ubiquitously expressed protein, and signs of cell toxicity in non-neuronal types have been reported (see, for example, [261,262]). Therefore, the use of ssODNs will have to be in parallel with drugs that produce a systemic effect, such as AMPK activators.

Mechanical interventions to treat HD have also been described, but they are out of the scope of this review. These and all other currently performed clinical trials regarding HD are regularly reviewed in the Huntington’s Disease Clinical Trials Corner Series [263,264,265,266,267].

### 4.3. Conclusions

As we discussed, oxidative stress is a main player in HD, and approaches to avoid it are apparent. However, the challenges in making them work are substantial due to a lack of knowledge. Therefore, there is an urgent need for more in-depth study of the sources of free radicals and their targets. The strong connection between oxidative stress and inflammation also requires more attention. For example, there is evidence that oxidative stress produced by neurons can switch on inflammation by inducing the activation of astrocytes and microglia. In turn, activated brain immune cells produce ROS, pushing forward this vicious cycle. Intervening to stop the signalling pathways that lead to activation of these events may also prevent oxidative stress in neurons.

Therefore, new strategies may need further refinement in order to induce deep changes in the progression of HD. We believe that future therapies will consist of newer, more sophisticated antioxidant drugs, together with other strategies aimed at addressing other issues related to neuronal death in HD, like inflammation. One example is metformin, a strong anti-inflammatory and antioxidant molecule that is also able to target unfolded mHtt by activation of autophagy. These strategies may be combined with genetic and cellular tools to prevent or reduce the expression of *HTT*, further reducing the source of all insults, mHTT. We believe that this combination of therapies could produce synergistic outcomes that could stop or delay the progression of this devastating disease.

## Figures and Tables

**Figure 1 antioxidants-09-00577-f001:**
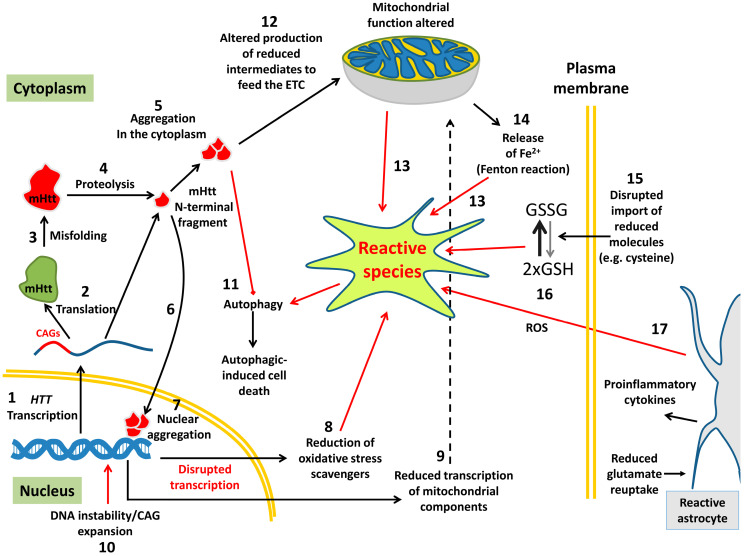
Mutant huntingtin (mHtt) disrupts a series of cellular processes that leads to the production of oxidative stress and produces a series of disruptions in important cellular processes. Once *HTT* is expressed (1), the messenger is processed through splicing and sent to the cytoplasm for translation (2). The mRNA encoding mHtt contains CAG triplets that can fold in secondary structures and sequester proteins (splicing factors and others), inducing cellular toxicity. Upon translation, mHtt can misfold (3) and become a substrate of proteases (4) which, after digestion, produce different fragments that are very toxic and prone to aggregation (5). These fragments can travel back to the nucleus (6), where they also aggregate (7). These nuclear aggregates can bind different transcription factors, including some genes encoding free radical scavengers (8) and components of the mitochondria (9), which enhances the toxic effects of oxidative stress. Free radicals alter DNA, which causes further expansions of the CAG tandems (10) when the cellular repair machinery opens the damaged DNA. In the cytoplasm, these aggregates interfere with autophagy (11) and the ubiquitin proteasome system (UPS). The aggregates are lipophilic, and therefore able to get between the membranes of the mitochondria (12), causing its malfunction, which in turn produces free radicals (13). Dysfunctional mitochondria liberate free iron (14), which can further induce the production of free radicals through the Fenton reaction. mHtt also causes defects in the plasma membrane (15), which cannot incorporate cysteine, a main component of the glutathione system, to the cytoplasm [14], hence glutathione cannot be reduced (16), enhancing the production of reactive species. Astrocytes respond to neuronal damage becoming activated in the form of reactive astrocytes. These cells produce proinflammatory cytokines and reactive oxygen species (ROS), among other deleterious events, inducing additional damage to neurons, further contributing to this vicious cycle.

**Figure 2 antioxidants-09-00577-f002:**
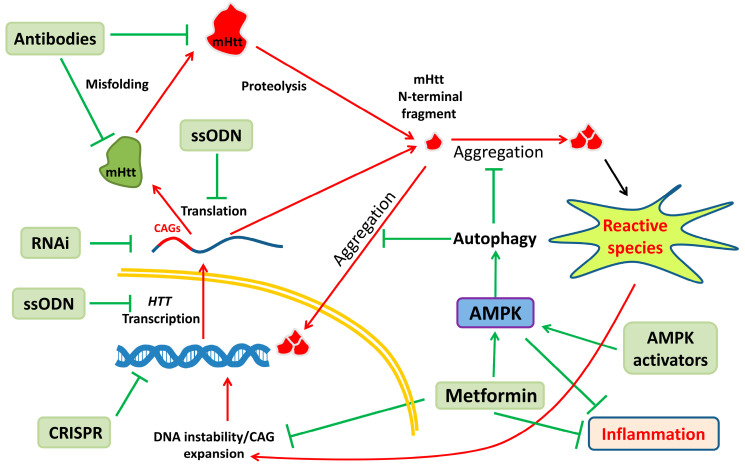
Strategies to reduce the expression of mutant huntingtin. The expression of mHtt causes a range of catastrophic events that disrupt neuronal functions, which can lead to cell death: gene expression disruption, abnormal expansions of CAG tandems, autophagy disruption, oxidative stress, etc. Many strategies to stop the progression of these events have been assayed, such as targeting oxidative stress by scavenging free radicals. However, the sources of reactive species remain, and further free radicals may be produced. We suggest that removing the cause of the production of these reactive species may be a more efficient way to treat HD. For example, ablation of the mutant allele of *HTT* by means of CRISPR will stop the production of mHtt, constituting a permanent cure for HD. This strategy lacks maturity; knocking down the gene using ssODN or RNAi may be a more practical approach. Antibodies against mHtt can be designed with ubiquitine ligase domains, and once introduced into cells, can target the toxic mHtt to degradation. Finally, mHtt can be degraded, taking advantage of the natural pathway of damaged protein clearance of the cell, autophagy. Activators of AMPK, a well-known autophagy inducer, may be considered as potential HD treatments, because they can reactivate autophagy in cells stressed by mHtt, which is ussually blocked in mouse models and patients with HD. The activation of AMPK results in reduced inflammation. Metformin has been shown to be a good candidate to reduce toxicity induced by mHtt. This drug is a regular treatment for type 2 diabetics, so its use has been widely investigated, and it has been shown to have few side effects. Metformin shows strong anti-inflammatory and antioxidant properties, AMPK-dependent and independent, and induces DNA stability, which may be useful to stop expansions of CAG tandems.

**Table 1 antioxidants-09-00577-t001:** Antioxidant therapies carried out in *Caenorhabditis elegans*
^1^.

polyQ Model ^2^	Compound	Effect	Reference
40Q::YFP in muscle cells (AM141)	α-lipoic acid	No effect in number of inclusion bodies	[115]
	*Cassia fistula* extract	Decreased number of inclusion bodies	[116]
	*Epigallocatechin gallate*	No effect in number of inclusion bodies	[115]
	*Epimedium brevicornum* acidic polysaccharide EbPS-A1	No effect in number of inclusion bodies	[117]
	Guarana extract	Decreased number of inclusion bodies	[118]
	Ruthenium (II) complexes	Decreased number of inclusion bodies	[119]
	Phycocyanin	Decreased number of inclusion bodies	[120]
	Salidroside	No effect in number of inclusion bodies	[121]
	Vitamin C	No effect in number of inclusion bodies	[115]
Htn150Q in ASH neurons (HA759)	*Dictiophora indusiata* polysaccharides	Attenuated polyQ-mediated chemosensory behaviour dysfunction	[122]
	*Epimedium brevicornum* acidic polysaccharide EbPS-A1	Increased survival rate under stress-induced conditions, reduced ROS levels and lipid peroxidation and increased SOD and CAT; attenuated polyQ-mediated chemosensory behaviour dysfunction	[117]
	Guarana extract	Attenuated polyQ-mediated chemosensory behaviour dysfunction	[123]
	Ruthenium (II) complexes	Increased neuronal survival and improved chemosensory behaviour	[119]
	Salidroside	Decreased lipid peroxidation and ROS and increased antioxidant enzyme activity; attenuated polyQ-mediated chemosensory behaviour dysfunction and increased neuronal survival	[121]

^1^ This table is not meant to be exhaustive. ^2^ AM141 strain expresses 40Q in muscle cells. HA759 strain encodes the N-terminus of human Htt carrying a polyQ tract (Htn150Q) fused to GFP and is expressed mainly in polymodal sensory ASH neurons.

**Table 2 antioxidants-09-00577-t002:** Antioxidant therapies carried out in *Drosophila melanogaster*
^1^.

HD Model ^2^	Compound	Effect	Reference
Htt-128Q	α-tocopherol and coenzyme Q10	No increase in survival rate and pupal mortality	[141]
Httex1p 93Q	Curcumin	Reduced neuronal loss in photoreceptors and improved polyQ-induced motor neuronal dysfunction	[150]
	Fisetin and resveratrol	Increased photoreceptor neuronal survival and rescued neuronal degeneration	[144,146]
	Green tea	Reduced mHtt-induced neurodegeneration, increased lifespan, inhibited aggregation, prevented photoreceptor degeneration and improved motor function	[152,153]
	Grape seed polyphenolic extract (GSPE)	Positive impact on longevity	[147]

^1^ This table is not meant to be exhaustive. ^2^ Htt-128Q flies express N-terminal 548 amino acids of Htt with 128Q and Httex1p 93Q model expresses exon 1 of Htt with 93Q.

**Table 3 antioxidants-09-00577-t003:** Antioxidant therapies carried out in rodent models of HD ^1^.

HD Model ^2^	Compound	Effect	Reference
R6/1 mice	*N*-Acetylcysteine	Reversed depressed-like behaviours, decreased oxidative damage markers in mitochondria and delayed onset of motor problems	[159,160]
	Linoleic acid, Q-linolenic acid, eicosapentaenoic acid, docosahexaenoic acid, K-lipoic acid and d-K-tocopherol	Improved survival rate and motor performance	[161]
R6/2 mice	α-lipoic acid	Modest improvement in survival	[162]
	Coenzyme Q10	Increased survival, delayed appearance of motor deficits, reduced body weight loss, cerebral atrophy and intranuclear inclusions of mHtt	[163]
	Creatine	Reduced mHtt aggregates, improved survival of animals and delayed mHtt-induced atrophy of striatal neurons	[164]
	Curcumin	Improved motor performance, activation of pro-survival pathways and maintenance of gastro-intestinal homeostasis	[165]
	Cystamine	Extended lifespan and reduced motor deficits	[166,167]
	Deferoxamine	Improved behavioural performance	[168]
	Dimethylfumarate	Extended lifespan, beneficial effects on motor behaviour and preserved neuronal morphology	[169]
	Grape seed polyphenolic extract (GSPE)	Increased lifespan and improved motor performance	[147]
	Melatonin	Decreased mortality rate and delayed HD onset	[170]
	XJB-5-131	Prevented body weight loss and improved motor performance	[171]
N171-82Q mice	Creatine	Improved survival rate, delayed motor dysfunction and body weight loss with reduced aggregated mHtt	[172]
	l-carnitine	Increased survival rate, reduced number of aggregates and improved motor performance	[173]
	Selenium derivatives	Reduced mHtt aggregates and decreased oxidized glutathione levels	[174]
YAC128 mice	Dimethylfumarate	Beneficial effects on motor behaviour and preserved neuronal morphology	[169]
	Ethyl-eicosapentaenoic acid	Modest improvement of motor function	[175]
	Resveratrol	Improved motor performance and learning	[176]
*Hdh*^(CAG)150/(CAG)150^ mice	XJB-5-131	Prevented body weight loss, mitochondrial dysfunction and oxidative damage, and improved motor performance	[177,178]
CAG140 knock-in mice	Curcumin	Ameliorated number of aggregates and partial improvement of motor function	[179]
3-NP-induced rats	α-lipoic acid	Decreased mitochondrial swelling and reduced cognitive impairment	[180]
	α-tocopherol and coenzyme Q10	Failed to achieve restoration of decline in mitochondrial production of energy	[181]
	FK-506	Prevented body weight loss, ameliorated locomotor activity, restored antioxidant enzyme activity and attenuated mitochondrial dysfunction	[182]
	Lycopene	Ameliorated mitochondrial function and prevented locomotor and memory deficits	[183]
	*N*-Acetylcysteine	Prevented mitochondrial dysfunction, behavioural deficits and neuronal death and attenuated lipid peroxidation	[184]
	Nicotinamide	Prevented increase in nitrite levels and malondialdehyde and depletion of GSH, improved motor performance and impeded neuronal death in striatum	[185]
	Quercetin	Reversed mitochondrial dysfunction, prevented decrease in antioxidant enzyme activity and improved motor performance; combination with lycopene alleviated anxiety and depression	[186,187]
	Rutin	Protected against body weight loss, restored antioxidant enzyme activity and improved motor and memory performance	[188]
	Selenium derivatives	Prevented body weight loss, induced better motor performance and restored CAT levels, reduced mHtt aggregates and decreased oxidized glutathione levels	[189]
	Virgin olive oil	Powerful brain antioxidant	[190]
Manolic acid-induced rats	α-tocopherol	Reversed body weight loss, disrupted motor coordination and oxidative stress by preserving mitochondrial function	[191]
Quinolinic acid-induced rats	Idebenone	Did not induce neuroprotection	[192]
	Sulforaphane	Prevented mitochondrial dysfunction	[193]

^1^ This table is not meant to be exhaustive. ^2^ R6/1 and R6/2 mice models express exon 1 of the *HTT* gene with 115 or 145 CAG repeats, respectively. N171-82Q mice express the N-terminal fragment of Htt with 82Q. YAC128 mice express a yeast artificial chromosome containing a full-length mHtt with 128Q. *Hdh*^(CAG)150/(CAG)150^ mice express 150 CAG in *Hdh*, the mouse HD gene homologue. CAG140 knock-in mice express a chimeric mouse/human exon 1 with 140 CAG repeats inserted in the murine huntingtin gene (*Hdh*). 3-nitropropionic acid (3-NP)-induced, manolic acid-induced and quinolinic acid-induced rats are transitory models produced by injections of chemicals.

**Table 4 antioxidants-09-00577-t004:** Clinical assays using antioxidants for therapeutic intervention in HD.

Compound	Mechanism	Study Design	No. of Patients (Test/Control)	Results	ClinicalTrials.Gov Identifier (Acronym)	Ref.
A-tocopherol (3000 IU daily)	Antioxidant	Double-blind, placebo-controlled	40/33	No effect on neurologic and neuropsychiatric symptoms Therapeutic effect on neurologic symptoms for patients in early HD	Not registered	[227]
Coenzyme Q10 (600 mg, 1200 mg, 2400 mg daily)	Rise in energy and antioxidant	Double blind, placebo-controlled	45/45	Well tolerated No significant slowing of functional decline in early HD	NCT00920699 (PREQUEL)	[228]
Coenzyme Q10 (2400 mg daily)	Rise in energy and antioxidant	Double blind, placebo-controlled	224/240	Futility analysis failed to show likelihood of benefit of CoQ 2400 mg/day	NCT00608881 (2CARE)	[229,239]
Creatine (8 g daily)	Rise in energy and antioxidant	Double blind, placebo-controlled	32/32	Restored regular levels of 8-OHdG in serum	Not registered	[100]
Creatine (30 g daily)	Rise in energy and antioxidant	Open-label	Not determined	Slowed ongoing cortical atrophy	Not registered	[231]
Creatine (10–30 g daily)	Rise in energy and antioxidant	Double blind, placebo-controlled	47/17	Safe and tolerable Neuroimaging demonstrated treatment-related slowing of cortical and striatal atrophy	NCT00592995 (PRECREST)	[230]
Creatine (40 g daily)	Rise in energy and antioxidant	Double blind, placebo-controlled	275/278	Creatine is not useful to treat HD	NCT00712426 CREST-E	[232]
Idebenone (90 mg 3 times a day)	Rise in energy and antioxidant	Double blind, placebo-controlled	50/50	No differences between groups; larger numbers needed for higher statistical power	Not registered	[233]
Fatty acids (8 g HUFAs daily)	Antioxidant	Double blind, placebo-controlled	10/7	Improved motor function	Not registered	[234]
Fatty acids (2 g ethyl-EPA daily)	Antioxidant	Pilot study, double blind, placebo-controlled	7/4	Improved motor function and MRI changes	Not registered	[235]
Fatty acids (2 g ethyl-EPA daily)	Antioxidant	Double blind, placebo-controlled	67/68	Improved motor function	Not registered	[236]
Fatty acids (1 g ethyl-EPA daily)	Antioxidant	Double blind, placebo-controlled	150/150	Ethyl-EPA was not beneficial	NCT00146211 TREND-HD	[237]
OPC-14117 (60 mg, 120 mg, 240 mg daily)	Antioxidant	Double blind, placebo-controlled	32/32	No effect	Not registered	[238]

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
