# Peer review of "Reactive Species in Huntington Disease: Are They Really the Radicals You Want to Catch?"

_antioxidants, 2020, doi:10.3390/antiox9070577_

Round 1

Reviewer 1 Report

The review by Dr. Yagie et al. describes the evidence about the involvement of oxidative stress in the pathogenesis of Huntington’s disease (HD) and address the results obtained by animal models as well as the negative results obtained by clinical trials. The topic is interesting, but there are some issues with the construction of the review that render it hard to read and to follow.

While I think that the section 4 is fluent, nicely written and interesting, together with part of section 3.4 regarding clinical trials on humans, the other section of the review are too long and dispersive. This make it difficult to focus on the main question of the review, which is if we really need to stop oxidative stress, or we should better focus first to target other impaired pathways in conjunction with oxidative stress. In fact, it is partially clear from the review that other pathways like UPS, autophagy and so on are strongly impaired by mHtt. So, it may seem that oxidative stress could be just a “collateral” event not playing the main role in the pathogenesis of the disease. Although proof of the involvement oxidative stress as pathological mechanism in HD exist, as correctly addressed by the authors the failing of clinical trials may point towards the other direction.

Therefore, I strongly suggest to shorten the first part of the review.

Here are some suggestions. The introduction is fine, whereas the parts 2.1 and 2.2 could be shortened a bit and demanded to other reviews for extensive explanations.

In my opinion, the section 2.3, since it is dealing with other neurological disorders, could be removed as it is not felt as instrumental to the HD as it may seems.

The section 3 from its beginning to 3.2 included are quite ok (see other minor issues below).

The section 3.3 could be re-organized by first describing the animal models used (so C. elegans, drosophila and mice/rats), with just the main alterations in oxidative stress markers. Then, there might be another section addressing only the antioxidant “therapies”. At this regard, the parts describing the results about the use of antioxidants for the treatment of HD in animal models could be also made more fluent and concise.

Finally, in section 3.3.3. you said that you will address the experimental animal trials within the clinical trial section. However, it seems not very appropriate and thus should be addressed in a separate section.

Following are also few minor issues.

- I suggest the review of a native English since some sentences are difficult to interpret. Also, read it carefully because there are some verbs and terms written incorrectly. For instance Line 94, line 313: “it’s” instead of “its”. Line 223: “Copper” instead of “Cooper”. Line 429: the beginning of the sentence seems truncated.

- In figure 1, consider to put numbers marking the key events (the same should be done in the legend) in order to make it easier to the proposed sequence of events leading to oxidative stress and other pathogenic key features.

At the end of Figure 1 legend, what do you mean for “mHtt causes defects in the plasma membrane, which cannot incorporate cysteine to the cytoplasm”? Do you mean glutathione? Anyway, I think that the right term is import, not incorporate.

- On line 196-197, you say that mHtt promotes cell death by mis-regulation of antioxidant gene expression. You should just add few sentences to discuss more in detail the mechanistic point of view.

- On line 227-228, you say that mHtt can reduce the expression of NfkB thus exacerbating oxidative stress. You should add few sentences describing the tentative mechanism.

Author Response

We thank very much the editor and the referees for their constructive suggestions and their positive comments. We write after each comment, to highlight our responses to each question of the referees.

Reviewer 1

The review by Dr. Yagie et al. describes the evidence about the involvement of oxidative stress in the pathogenesis of Huntington’s disease (HD) and address the results obtained by animal models as well as the negative results obtained by clinical trials. The topic is interesting, but there are some issues with the construction of the review that render it hard to read and to follow.

While I think that the section 4 is fluent, nicely written and interesting, together with part of section 3.4 regarding clinical trials on humans, the other section of the review are too long and dispersive. This make it difficult to focus on the main question of the review, which is if we really need to stop oxidative stress, or we should better focus first to target other impaired pathways in conjunction with oxidative stress. In fact, it is partially clear from the review that other pathways like UPS, autophagy and so on are strongly impaired by mHtt. So, it may seem that oxidative stress could be just a “collateral” event not playing the main role in the pathogenesis of the disease. Although proof of the involvement oxidative stress as pathological mechanism in HD exist, as correctly addressed by the authors the failing of clinical trials may point towards the other direction.

We agree with the reviewer that some bits of the manuscript may gain fluidity and clarity with some shortening. We have proceed accordingly (see below).

Therefore, I strongly suggest to shorten the first part of the review.

Here are some suggestions. The introduction is fine, whereas the parts 2.1 and 2.2 could be shortened a bit and demanded to other reviews for extensive explanations.

In my opinion, the section 2.3, since it is dealing with other neurological disorders, could be removed as it is not felt as instrumental to the HD as it may seems.

We have rearranged the whole point 2. Now this section is called “Oxidative stress”. We have shortened sections 2.1 and 2.3., as suggested by the referee, and we have deleted the whole section 2.3.

The section 3 from its beginning to 3.2 included are quite ok (see other minor issues below).

Many thanks to the referee for their positive comments.

The section 3.3 could be re-organized by first describing the animal models used (so C. elegans, drosophila and mice/rats), with just the main alterations in oxidative stress markers. Then, there might be another section addressing only the antioxidant “therapies”. At this regard, the parts describing the results about the use of antioxidants for the treatment of HD in animal models could be also made more fluent and concise.

We have done accordingly. Now all animal models are explained at the beginning of each section (worms, flies and murine models of HD), including their alterations in oxidative stress, whenever there is data about them.

Finally, in section 3.3.3. you said that you will address the experimental animal trials within the clinical trial section. However, it seems not very appropriate and thus should be addressed in a separate section.

We have moved all animal data to the end of the previous section, pointing out that these were the basis of the clinical trials described below, as suggested by the referee.

 Following are also few minor issues.

- I suggest the review of a native English since some sentences are difficult to interpret. Also, read it carefully because there are some verbs and terms written incorrectly. For instance Line 94, line 313: “it’s” instead of “its”. Line 223: “Copper” instead of “Cooper”. Line 429: the beginning of the sentence seems truncated.

We thank the referee for spotting these mistakes. We have corrected every point described above, and also we have sent the paper for appropriate English editing to the services of the MDPI editorial.

- In figure 1, consider to put numbers marking the key events (the same should be done in the legend) in order to make it easier to the proposed sequence of events leading to oxidative stress and other pathogenic key features.

We agree with the referee that this may help to follow the description of the figure. So we have done these changes accordingly, explaining the figure through points 1 to 17.

At the end of Figure 1 legend, what do you mean for “mHtt causes defects in the plasma membrane, which cannot incorporate cysteine to the cytoplasm”? Do you mean glutathione? Anyway, I think that the right term is import, not incorporate.

We really mean cysteine. Cysteine is part of the core of the glutathione system, and mutant huntingtin is actually affecting plasma membrane in terms of cysteine permeability. We have rephrased this a bit, to make it clearer, and added a reference.

- On line 196-197, you say that mHtt promotes cell death by mis-regulation of antioxidant gene expression. You should just add few sentences to discuss more in detail the mechanistic point of view.

We agree with the referee, we didn’t explain this well in detail. We have rephrased the paragraph, to explain that one of the pathways misregulated, CREB-PCG1alpha, regulates components of the mitochondrial antioxidant defence. We also have added a new citation to point this out.

- On line 227-228, you say that mHtt can reduce the expression of NfkB thus exacerbating oxidative stress. You should add few sentences describing the tentative mechanism.

Again we agree with the referee, and we thank him/her for her insights. We have extended the explanation around the signalling events that lead to reduced expression, linked to mHtt-induced reduction of NfkB, which leads to reduced antioxidant capacity and cell death.

Reviewer 2 Report

The standard of English is not high enough for a fair scientific review

Author Response

We thank very much the editor and the referees for their constructive suggestions and their positive comments. We writeafter each comment, to highlight our responses to each question of the referees.

Reviewer 2

The standard of English is not high enough for a fair scientific review

We are not native English speakers, and therefore we must agree with the reviewer. The third referee, though, do not think our English is so bad. In any case, we requested to the English editing services from MDPI, to perform an extensive English editing.

Reviewer 3 Report

The report “Free radicals in Huntington disease: are they really the extremists you want to arrest?” by Bono-Yagüe et al., is well written with suitable  English and results, figures are good enough and the review is interested to the field, but with major revision.

Major revision:

  1. What happen with glia actions in Huntington disease? This point should be determined. The number of astrocytes inside the brain are higher than microglia and they are the first line of defence, please add more information about it.
  2. Changes in microglia action after increment of oxidative stress will be indicated. Remember that microglia action will revert quickly after injury.
  3. Indicate the changes in the astrocytes detected in Huntington disease. The changes in number, their actions and the types of astrocytes during development of the disease.
  4. The conclusion are not strong enough. Authors need to improve it. This will be better to the authors and to the journal.
  5. What happen with inflammation? In many neurodegenerative disease such as AD, ALS and MS, inflammation play an important mechanisms, sometime in collaboration with oxidative stress. This will be a strong point to add to the review. Also in inflammation and oxidative stress the relationship between cells are important. Perhaps indicating the relationship between astrocytes and neurons, microglia and astrocytes, and microglia and neurons, will be important to determine their conclusions.

Minor revision:

  1. Some errors are detected in the English redaction.
  2. Bibliography is good enough but probably of a larger extent would be desirable.

Author Response

We thank very much the editor and the referees for their constructive suggestions and their positive comments. We write after each comment, to highlight our responses to each question of the referees.

Reviewer 3

The report “Free radicals in Huntington disease: are they really the extremists you want to arrest?” by Bono-Yagüe et al., is well written with suitable  English and results, figures are good enough and the review is interested to the field, but with major revision.

We thank very much the referee for this kind description of the paper.

Major revision:

What happen with glia actions in Huntington disease? This point should be determined. The number of astrocytes inside the brain are higher than microglia and they are the first line of defence, please add more information about it.

Changes in microglia action after increment of oxidative stress will be indicated. Remember that microglia action will revert quickly after injury.

Indicate the changes in the astrocytes detected in Huntington disease. The changes in number, their actions and the types of astrocytes during development of the disease.

We agree with the referee. Oxidative stress and inflammation play key roles during the progression of HD. We did not include it for the sake of simplicity and because the Special Issue of Antioxidants was specifically dedicated to “oxidative stress and Rare Diseases”. Moreover, inflammation is a vast field of research within the neurodegenerative diseases field, so we were uneasy to add so much more information about it. However, we hope that the referee and the editor agree with us that we wrote in a way that to kept the review simple, yet making the message of the paper stronger.

In summary, we have added a substantial amount of information and discussion about inflammation. We have added a small section devoted to explain the role of inflammation in HD, and the relationship between oxidative stress and inflammation (3.2. –Oxidative stress and neuroinflammation in HD). We have explained the changes that activation induces to microglia and astrocytes. We have updated Figures 1 and 2, to accommodate this issue. In Figure 1 we represent a reactive astrocyte cell, secreting pro-inflammatory molecules and ROS. In Figure 2 we point out that AMPK and metformin are anti-inflammatory agents.

The conclusion are not strong enough. Authors need to improve it. This will be better to the authors and to the journal.

What happen with inflammation? In many neurodegenerative disease such as AD, ALS and MS, inflammation play an important mechanisms, sometime in collaboration with oxidative stress. This will be a strong point to add to the review. Also in inflammation and oxidative stress the relationship between cells are important. Perhaps indicating the relationship between astrocytes and neurons, microglia and astrocytes, and microglia and neurons, will be important to determine their conclusions.

We have added some phrases in conclusion to include the issue regarding inflammation and oxidative stress, in the context of HD.

We have also updated the abstract and the keywords, to mention the issue regarding microglia, astrocytes and inflammation.

Minor revision:

Some errors are detected in the English redaction.

As stated above, we have requested the editorial system to perform a deep English editing of the manuscript.

Bibliography is good enough but probably of a larger extent would be desirable.

We have added some new citations regarding inflammation, astrocyte activation, etc., as requested by the referee.

Round 2

Reviewer 1 Report

I think that the authors correctly replied to my concerns. 

Author Response

We thank the reviewer for his/her kind comments

Reviewer 2 Report

With the suggested added sections and some rearrangements, alongside the Publisher's English correction, this manuscript is suitable for publication. The use of tables to summarse the key points is helpful and the citations are suitably up-to-date.

I have, if the authors do not mind, some minor suggestions for the abstract to improve the visibility of the manuscript amongst the many reviews on HD pathogenesis and clinical approches:

Huntington disease (HD) is a rare neurodegenerative condition affecting between 1 and 10 people in every 100,000 in western countries. The causative gene, HTT, encodes huntingtin, a protein of as-yet unknown function. Mutant huntingtin causes a range of cellular phenotypes, including oxidative stress and the activation of microglia and astrocytes, which in turn leads to chronic inflammation of the brain. Although substantial efforts have been made to find a cure for HD, there are currently no medical interventions that are able to prevent or even delay the progression of the disease. Among the many potential therapeutic strategies proposed, oxidative stress and inflammation have been extensively studied and clinical trials targeting these specific pathways have been undertaken. Here we review the range of research investigating the importance of oxidative stress in HD and the approaches used to counteract this potentially damaging process. Many of these protective strategies have produced positive results, yet no substantial functional recovery has been observed in animal models or in HD patients to date. We discuss possible explanations for this issue and suggest potential ways to improve clinical efficacy of oxidative stress-associated therapies.

My last main concern is still the title - the use of 'extemists' and 'arrest' is highly unsuitable and should be changed.

Author Response

We thank the reviewer for his/her comments.

We have changed these words, as suggested by the referee. Now the title of the paper is as follows:

Reactive species in Hutington disease: are they really the radicals you want to catch?

Reviewer 3 Report

Accept

Author Response

We thank the reviewer for his/her work and nice comments.